# Validated Preclinical Mouse Model for Therapeutic Testing against Multidrug-Resistant *Pseudomonas aeruginosa* Strains

Jonathan M. Warawa,[a,b] Xiaoxian Duan,[a] Charles D. Anderson,[a] Julie B. Sotsky,[a] Daniel E. Cramer,[a] Tia L. Pfeffer,[a] Haixun Guo,[a,c] Scott Adcock,[a] Alexander J. Lepak,[d] David R. Andes,[d] Stacey A. Slone,[e] Arnold J. Stromberg,[e] Jon D. Gabbard,[a] William E. Severson,[a] Matthew B. Lawrenz[a,b]

aCenter for Predictive Medicine for Biodefense and Emerging Infectious Diseases, University of Louisville, Louisville, Kentucky, USA

bDepartment of Microbiology and Immunology, University of Louisville School of Medicine, Louisville, Kentucky, USA

cDepartment of Radiology, University of Louisville School of Medicine, Louisville, Kentucky, USA

dDepartment of Medicine, University of Wisconsin, Madison, Wisconsin, USA

eDr. Bing Zhang Department of Statistics, University of Kentucky, Lexington, Kentucky, USA

**ABSTRACT** The rise in infections caused by antibiotic-resistant bacteria is outpacing the development of new antibiotics. The ESKAPE pathogens (*Enterococcus faecium, Staphylococcus aureus, Klebsiella pneumoniae, Acinetobacter baumannii, Pseudomonas aeruginosa*, and *Enterobacter* species) are a group of clinically important bacteria that have developed resistance to multiple antibiotics and are commonly referred to as multidrug resistant (MDR). The medical and research communities have recognized that, without new antimicrobials, infections by MDR bacteria will soon become a leading cause of morbidity and death. Therefore, there is an ever-growing need to expedite the development of novel antimicrobials to combat these infections. Toward this end, we set out to refine an existing mouse model of pulmonary *Pseudomonas aeruginosa* infection to generate a robust preclinical tool that can be used to rapidly and accurately predict novel antimicrobial efficacy. This refinement was achieved by characterizing the virulence of a panel of genetically diverse MDR *P. aeruginosa* strains in this model, by both 50% lethal dose ($LD_{50}$) analysis and natural history studies. Further, we defined two antibiotic regimens (aztreonam and amikacin) that can be used as comparators during the future evaluation of novel antimicrobials, and we confirmed that the model can effectively differentiate between successful and unsuccessful treatments, as predicted by *in vitro* inhibitory data. This validated model represents an important tool in our arsenal to develop new therapies to combat MDR *P. aeruginosa* strains, with the ability to provide rapid preclinical evaluation of novel antimicrobials and support data from clinical studies during the investigational drug development process.

**IMPORTANCE** The prevalence of antibiotic resistance among bacterial pathogens is a growing problem that necessitates the development of new antibiotics. Preclinical animal models are important tools to facilitate and speed the development of novel antimicrobials. Successful outcomes in animal models not only justify progression of new drugs into human clinical trials but also can support FDA decisions if clinical trial sizes are small due to a small population of infections with specific drug-resistant strains. However, in both cases the preclinical animal model needs to be well characterized and provide robust and reproducible data. Toward this goal, we have refined an existing mouse model to better predict the efficacy of novel antibiotics. This improved model provides an important tool to better predict the clinical success of new antibiotics.

**KEYWORDS** *Pseudomonas aeruginosa*, preclinical models, antibiotic testing, mouse models, pharmacokinetics

Address correspondence to Matthew B. Lawrenz, matt.lawrenz@louisville.edu.

The authors declare no conflict of interest.

It is estimated that ≥1,000,000 people worldwide die each year due to infections caused by antibiotic-resistant bacteria (1). Based on the current rate of antibiotic resistance observed in the clinic and without a concerted effort to identify and develop new antibiotics, the United Nations Interagency Coordinating Group on Antimicrobial Resistance predicted that deaths due to antibiotic-resistant bacteria could increase by several million by 2050 (2). The ESKAPE pathogens (*Enterococcus faecium*, *Staphylococcus aureus*, *Klebsiella pneumoniae*, *Acinetobacter baumannii*, *Pseudomonas aeruginosa*, and *Enterobacter* species) have been recognized by the Centers for Disease Control and Prevention (CDC) and the World Health Organization (WHO) as the leading causes of nosocomial infections throughout the world and have all acquired resistance to multiple antibiotics (3). *P. aeruginosa* is an opportunistic pathogen that can cause infection of the skin, eyes, urinary tract, and lungs and is especially pathogenic for immuno-compromised individuals. On its own, *P. aeruginosa* is estimated to be responsible for up to 20% of nosocomial infections in intensive care units in the United States and the European Union, with >35% of isolates being resistant to multiple antibiotics (4–6). Infections with multi-drug-resistant (MDR) *P. aeruginosa* strains are associated with significant increases in mortality rates, particularly among patients with respiratory-associated bacteremia (7).

To combat these infections, we need to identify and develop new antimicrobials against the ESKAPE pathogens. Toward this goal is the development of robust preclinical models that can quickly and accurately predict the efficacy of novel antimicrobials in humans. The mouse is the most widely used model for preclinical testing of antimicrobial investigational drugs (INDs). The main driving factors for this are (i) a large amount of data available on the pathogenesis of microbes in the mouse, (ii) the low costs to purchase, breed, and house mice, and (iii) the ability for researchers to incorporate increased group sizes and statistical power to overcome the experimental noise that is often associated with preclinical models. Experimental noise is considered a major contributor to past failures in translation of preclinical results into the clinic (8). Therefore, this third driving factor is critically important for successful preclinical testing. Importantly, the application of pharmacokinetic (PK)/pharmacodynamic (PD) analysis of antimicrobials in mice allows for the identification of driver indices that are independent of host metabolism and can be used to bridge preclinical data to human application (9). Therefore, preclinical data from mice on the efficacy of antimicrobials are significantly more predictive of success in the clinic than are data that have been reported for other classes of drugs. However, for the mouse to be an effective preclinical model, it should be well characterized to recapitulate human disease and validated to respond to current antimicrobials as expected with a variety of strains representing the potential diversity of isolates seen in the clinic.

We previously developed a lethal respiratory model of MDR *P. aeruginosa* infection in leukopenic BALB/c mice for the purpose of testing novel therapeutics (10). This model was chosen because it has been well characterized to mimic human disease, with bacterial proliferation in the lungs, acute inflammation, and development of pneumonia (10–12). Using this model, we have successfully evaluated the efficacy of numerous therapies either as monotherapy or as combination therapy with a representative carbapenem antibiotic, namely, meropenem (10, 13, 14). However, this model has limitations, including characterization with only one strain of *P. aeruginosa* and a lack of robust comparative antibiotics to gauge the success of novel INDs. Therefore, to improve our current model for preclinical screening of antimicrobials against *P. aeruginosa*, we validated our infection model with four additional strains of MDR *P. aeruginosa*, better representing the genetic diversity of clinical isolates. We also fully characterized two antibiotics under success and failure scenarios so that they can be used as comparator drugs to gauge the efficacy of novel antimicrobials against acute pulmonary infection with *P. aeruginosa*.

## RESULTS

**Identification of the lethal dose 50 of *P. aeruginosa* strains in the leukopenic mouse model.** Our first step in enhancing the predictive ability of the mouse model we developed previously for preclinical screening of antimicrobials against *P. aeruginosa* infection (10) was to characterize the virulence of different strains of MDR *P. aeruginosa* in this model. Toward this goal, we chose to characterize four *P. aeruginosa* strains that are

**TABLE 1** *P. aeruginosa* strains characterized in the mouse model

| Strain | Phenotype[a] Resistant | Sensitive | Intermediate | Known resistance[b] | BioSample accession number | LD$_{50}$ (CFU)[c] |
|---|---|---|---|---|---|---|
| 0230 | Ami, Cefe, Ceft, Cip, Dor, Gen, Imi, Lev, Mer, Pip, Tob | Azt, Col, Pol | | VIM-2 | SAMN04901620 | 20 |
| 0231 | Azt, Cefe, Ceft, Cip, Dor, Gen, Imi, Lev, Mer, Pip, Tob | Ami, Col, Pol | | KPC-5 | SAMN04901621 | 39 |
| 0241 | Cefe, Ceft, Cip, Dor, Gen, Imi, Lev, Mer, Pip, Tob | Col, Pol | Ami, Azt | IMP-1 | SAMN04901631 | 525 |
| 0246 | Ami, Azt, Cefe, Ceft, Cip, Dor, Gen, Imi, Lev, Mer, Pip, Tob | Col, Pol | | NDM-1 | SAMN04901636 | $4.07 \times 10^5$ |

[a]Ami, amikacin; Azt, aztreonam; Cefe, cefepime; Ceft, ceftazidime; Cip, ciprofloxacin; Col, colistin; Dor, doripenem; Gen, gentamicin; Imi, imipenem; Lev, levofloxacin; Mer, meropenem; Pip, piperacillin-tazobactam; Pol, polymyxin B; Tob, tobramycin.
[b]Genes associated with antibiotic resistance.
[c]LD$_{50}$ values calculated from these studies.

widely available to the research community through the CDC and Food and Drug Administration (FDA) Antimicrobial Resistance Isolate Bank *P. aeruginosa* panel (15). From this panel, the strains were chosen because their genomes have been sequenced, they have different antibiotic resistance profiles, including different known mechanisms of resistance to β-lactams, and they display different degrees of resistance to amikacin and aztreonam, which can be used as comparators in future studies (Table 1).

Because no virulence data are available for the isolates in the FDA-CDC Antimicrobial Resistance Isolate Bank *P. aeruginosa* panel, we first sought to identify the number of bacteria required to cause lethal infection in 50% of infected animals (i.e., the 50% lethal dose [LD$_{50}$]) for each strain in mice with cyclophosphamide-induced leukopenia (10). Male and female mice were infected by direct lung instillation with escalating doses of the four MDR *P. aeruginosa* strains (10, 16). Mice were monitored for the development of moribund disease at 8-h intervals. Mice that met predefined endpoint criteria were humanely euthanized and scored as succumbing to disease 8 h later. All four strains were able to establish lethal infection in leukopenic mice, but we observed differences in the LD$_{50}$ values between the strains. Strains 0230 and 0231 were the most virulent in the mice, with mortality rates of >50% of animals within 50 h postinfection achieved with a challenge of only $10^2$ CFU (Fig. 1A and B). The 0246 strain was slightly less virulent, as >50% of animals succumbed to infection at the dose of $10^3$ CFU (Fig. 1C). In contrast, strain 0241 was significantly less virulent than the other three strains, requiring $10^6$ CFU to achieve a mortality rate of >50% (Fig. 1D). For all strains tested, sex did not have a significant impact on the resistance of the host to lethal infection. Finally, at inocula resulting in mortality rates of ~50%, we observed similar mean times to death for all strains, ranging from 44 to 56 h. From these survival curves, we calculated the LD$_{50}$ values for strains 0230, 0231, 0246, and 0241 as 1.30, 1.59, 2.78, and 5.61 log CFU, respectively.

**Natural history of infection of *P. aeruginosa* strains in the leukopenic mouse model.** Once we established that each strain was able to cause lethal infection, we next sought to characterize the natural history of the infection in the leukopenic mouse model. Male and female mice were infected by direct instillation into the lungs with 10× the LD$_{50}$ of each of the *P. aeruginosa* strains. At 3, 6, 12, and 21 h, host temperatures were measured, and subsets of mice were humanely euthanized to harvest tissues for bacterial enumeration and histological analysis. Twenty-one hours was chosen as the last time point to minimize the number of animals reaching endpoint criteria prior to a designated time point, resulting in a need to exclude such animals from subsequent analyses. In general, mice that received the more virulent strains 0230, 0231, and 0246 and were inoculated with 3 to 4 log CFU of bacteria followed very similar courses of disease. At 3 h postinfection, ~3.0 log CFU of bacteria were recovered from the lungs of infected mice. Similar numbers were recovered at 6 h postinfection, but bacterial numbers increased by ~1.5 log CFU by 12 h postinfection. By 21 h postinfection, mean bacterial numbers were >6 log CFU for all three strains (Fig. 2). Bacterial dissemination from the lungs to the blood, as indicated by spleen colonization, was not detected prior to 21 h for any of the three strains (Fig. 2). Significant changes in the host response to infection, measured by hypothermia and lung inflammation, were

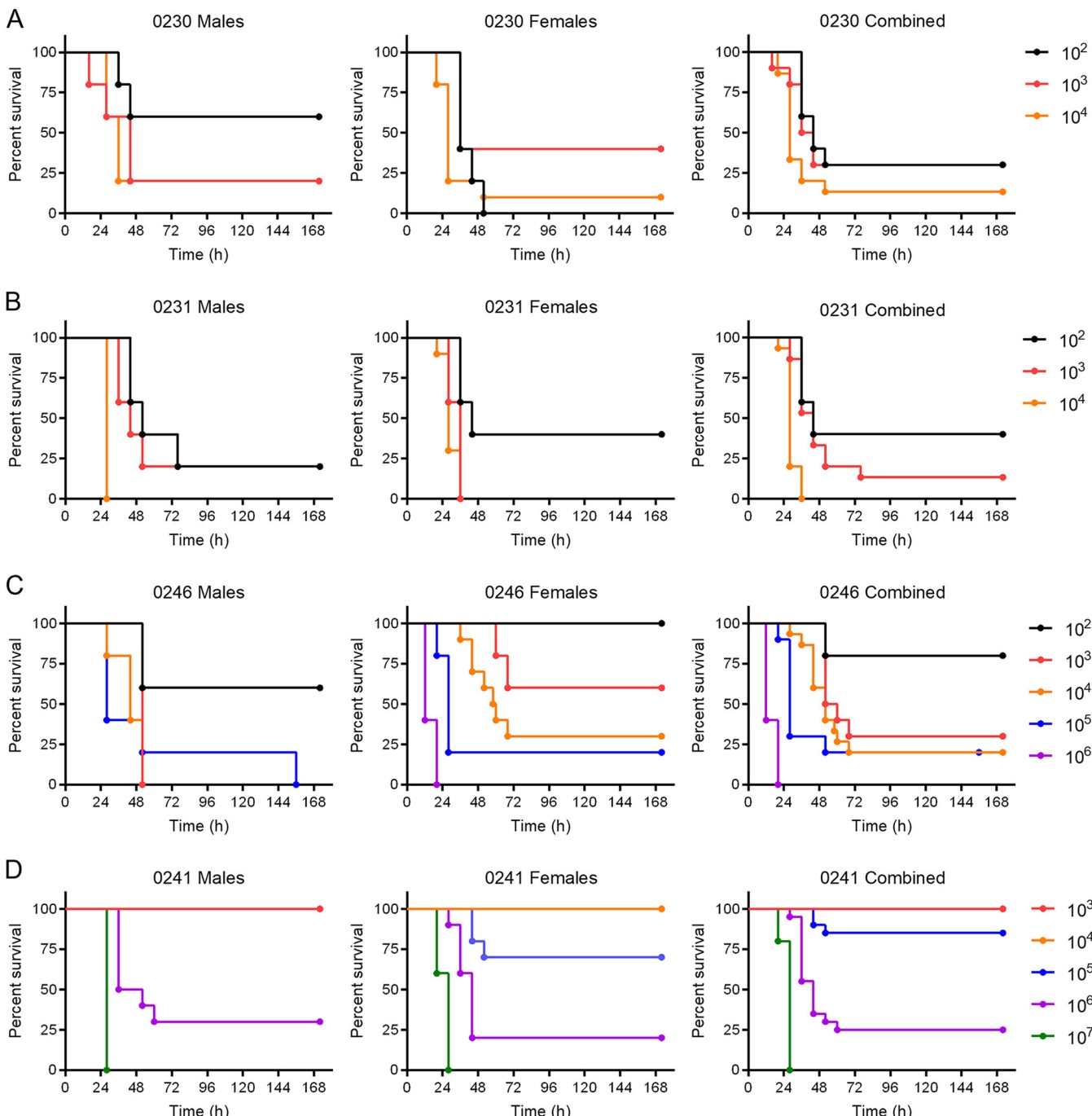

**FIG 1** Survival of mice infected with different strains of *P. aeruginosa*. Equal numbers of male and female mice (*n* = 10) were infected with the indicated CFU of *P. aeruginosa* strain 0230 (A), 0231 (B), 0246 (C), or 0241 (D) and monitored q8h for the development of moribund disease. Animals that met predetermined endpoint criteria were scored as succumbing to infection 8 h later.

also not observed in 0230-, 0231-, and 0246-infected mice until 21 h postinfection (Fig. 3).

Strain 0241, being less virulent than the other three strains, required a significantly larger inoculum to establish lethal disease (10× LD$_{50}$ = 6.6 log CFU). Therefore, significantly higher bacterial numbers were recovered from the lungs by 3 h postinfection (Fig. 2D). Bacterial burdens did not change over the 21-h period of observation. However, the elevated bacterial numbers in the lungs resulted in bacterial dissemination to the bloodstream and spleen within 3 h after inoculation (Fig. 2H). This earlier dissemination directly correlated with earlier development of hypothermia in the

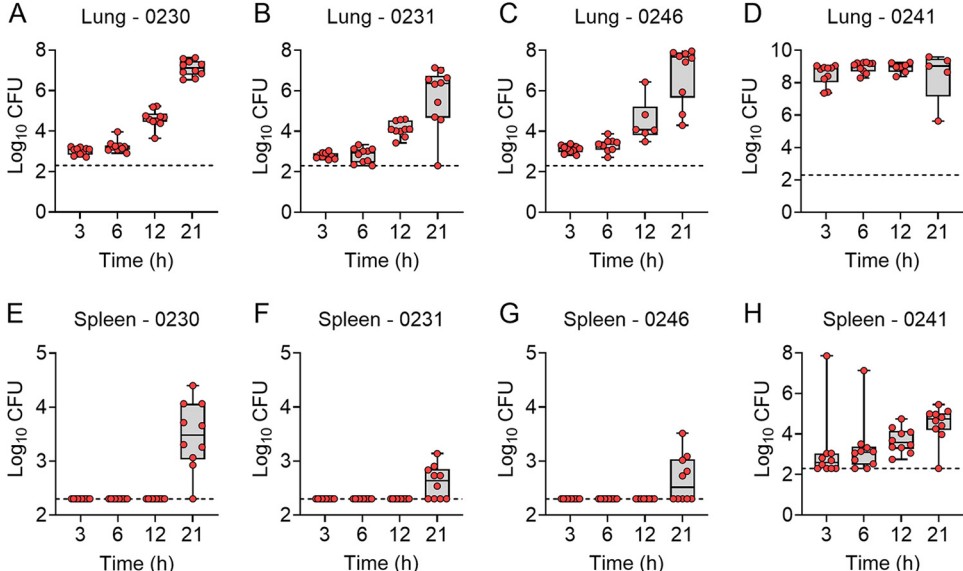

**FIG 2** Bacterial enumeration from tissues of infected mice. Equal numbers of male and female mice ($n = 10$) were infected with $10\times$ the $LD_{50}$ of the indicated *P. aeruginosa* strains. At 3, 6, 12, and 21 h, mice were euthanized, and bacterial numbers were enumerated from the lungs (A to D) or spleens (E to H). Each circle represents data from an individual mouse. The dotted line indicates the limit of detection.

animals infected with strain 0241 (Fig. 3D). However, despite increased bacterial burdens at earlier time points, lung inflammation and pathology were still not observed at 21 h postinfection (Fig. 3H). Together, these data established that the leukopenic mouse model is amenable to infection by a variety of different MDR *P. aeruginosa* strains and established parameters of infection that can be used to measure subsequent drug efficacy, including bacterial replication, dissemination, onset of hypothermia, and lung pathology.

**PK characterization of comparator antibiotics aztreonam and amikacin.** Our second goal in improving the predictive ability of the mouse model for preclinical screening of antimicrobials against *P. aeruginosa* infection was to establish an antibiotic that could be used as a comparator to gauge the efficacy of novel INDs. One of the criteria used during the selection of the panel of *P. aeruginosa* clinical isolates characterized above was differential susceptibility to an antibiotic, which would allow us to demonstrate both success and failure scenarios in the model. Therefore, strains 0230, 0231, 0246, and 0241 have different susceptibilities to two antibiotics, namely, aztreonam and amikacin (Table 1). Based on these phenotypes, we next sought to establish PK parameters for both antibiotics in the leukopenic mouse model during *P. aeruginosa* infection. Male and female mice were infected by direct instillation into the lung with $10\times$ the $LD_{50}$ of either strain 0230 (for mice receiving aztreonam) or strain 0231 (for mice receiving amikacin). Mice received escalating doses of antibiotics at 3 h postinfection, and plasma was collected at 5, 10, 15, 30, 60, and 120 min postantibiotic administration. Antibiotic concentrations in plasma were quantified and analytically assessed by high-performance liquid chromatography (HPLC) (Fig. 4 and Tables 2 and 3). PK constants were calculated using a time-ordered data set generated from the pooled PK data.

Using these data, we calculated dosing regimens for mice that would closely mimic the parameters achieved in the clinic for humans. For aztreonam, the time the drug concentration is above the MIC ($T_{>MIC}$) is the primary driver of success. Using human clinical data based on administration of 2 g of aztreonam intravenously every 8 h (q8h) as our target (17), we calculated the $T_{>MIC}$ values for mice receiving various regimens of aztreonam via subcutaneous (SC) injection (Table 4). Based on our PK data, 640 mg/kg/6 h SC in the mouse is predicted to achieve a free maximum drug concentration ($C_{max}$) value slightly higher than that in humans but would achieve $T_{>MIC}$ exposures against organisms (susceptible and resistant) that would be predicted to be therapeutically successful for a

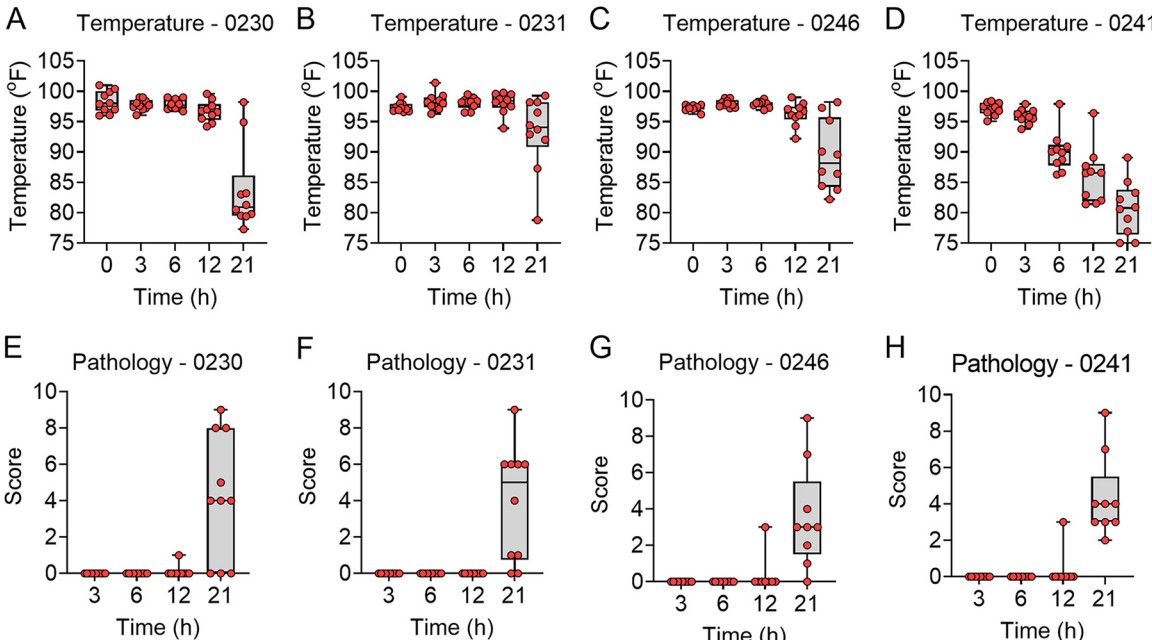

**FIG 3** Host responses during *P. aeruginosa* infection. Equal numbers of male and female mice (*n* = 10) were infected with 10× the LD$_{50}$ of the indicated *P. aeruginosa* strains. (A to D) Temperatures were measured at the indicated times. (E to H) At 3, 6, 12, and 21 h, mice were euthanized, a cross section of the lungs was excised and stained with H&E, and inflammation and pathology were assessed blindly. Each circle represents data from an individual mouse.

susceptible organism (MIC of 4 $\mu$g/mL), marginally successful for a resistant strain (MIC of 32 $\mu$g/mL), and unsuccessful for a highly resistant strain (MIC of >64 $\mu$g/mL).

For amikacin, the area under the concentration-time curve from 0 to 24 h (AUC$_{0-24}$) is the primary driver of success. Using human clinical data based on administration of 15 mg/kg amikacin to patients every 24 h (q24h) as our target (18, 19), we calculated dosing regimens for amikacin via SC injection in mice that matched the AUC$_{0-24}$ and $C_{max}$ values seen in human patients (Table 5). Treating mice q24h with a dose of 605 mg/kg would match the AUC$_{0-24}$ but exceed the $C_{max}$ by 8.3-fold. Similarly, treating mice every 12 h (q12h) with a dose of 385 mg/kg would match the AUC$_{0-24}$ but exceed the $C_{max}$ by 3.6-fold. Finally, treating mice every 6 h (q6h) with a dose of 293 mg/kg would match the AUC$_{0-24}$ but exceed the $C_{max}$ by only 1.7-fold. Based on these data, we predicted that drug administration regimens of 640 mg/kg/6 h for aztreonam and 293 mg/kg/6 h for amikacin would best represent a humanized dose of each drug in the leukopenic mouse model.

**Validation of humanized doses of antibiotics by bacterial burden reduction.** Having identified humanized dosing regimens for aztreonam and amikacin, we next sought to validate these regimens by demonstrating both success and failure scenarios in the leukopenic mouse model. Toward this goal, equal numbers of male and female mice were infected by direct instillation into the lungs with 10× the LD$_{50}$ of *P. aeruginosa* strain 0230 (sensitive to aztreonam and resistant to amikacin) or strain 0231 (resistant to aztreonam and sensitive to amikacin). At 6 h postinfection, mice were administered 640 mg/kg aztreonam, 293 mg/kg amikacin, or vehicle only (phosphate-buffered saline [PBS]). Drug administration was repeated q6h. At each administration, mouse temperature was also measured. At 21 h, mice were euthanized, and tissues were harvested for bacterial enumeration and histological analysis. Results were compared to tissues harvested from a subset of mice 6 h postinfection, representing the baseline values prior to the first administration of antibiotic.

As predicted by the *in vitro* MIC determinations, mice infected with *P. aeruginosa* 0230 (sensitive to aztreonam and resistant to amikacin) began developing signs of infection, including hypothermia, at 12 h postinfection when they were treated with PBS or amikacin but not when they received aztreonam (Fig. 5A). While low numbers of bacteria were recovered from

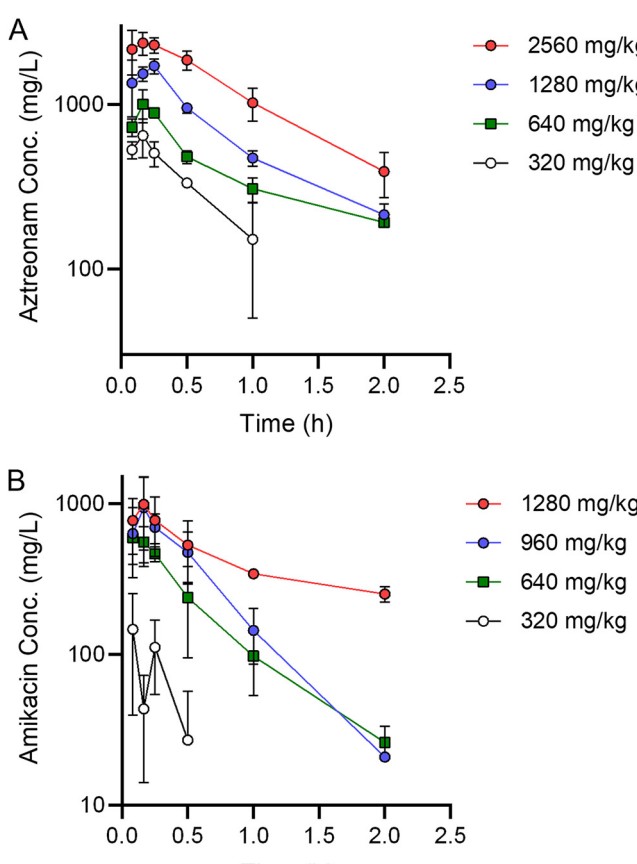

**FIG 4** Concentrations of antibiotics in plasma. Equal numbers of male and female mice (*n* = 4) were administered a single dose of aztreonam (A) or amikacin (B), and plasma drug concentrations were determined at the indicted time points. Doses are indicated in the keys. Each symbol represents the mean ± SD.

the lungs in some of the aztreonam-treated animals, the bacterial burden was significantly higher in the lungs of the PBS- and amikacin-treated mice, representing a difference of >3.0 log CFU between the groups (*P* < 0.0001) (Fig. 5B). Moreover, no bacteria were recovered from the spleens of aztreonam-treated mice, while mice treated with PBS or amikacin had disseminated infection, represented by significantly higher bacterial burdens in the spleens (*P* < 0.01) (Fig. 5C). Finally, while mild lung pathology was observed in aztreonam-treated mice, pathology scores were significantly higher in animals treated with PBS or amikacin (Fig. 5D). Similar trends were observed for mice infected with *P. aeruginosa* 0231 (resistant to aztreonam and sensitive to amikacin) except that disease severity, bacterial burden, and lung pathology were lower when mice were treated with amikacin, to which strain 0231 is sensitive (Fig. 6). Together, these data validate that the humanized dosing regimens for aztreonam and amikacin in this model of pulmonary *P. aeruginosa* infection mimic findings expected in the clinic, providing significant reductions in bacterial proliferation and dissemi-

**TABLE 2** Aztreonam plasma concentrations over time

| Time point (min) | Aztreonam concentration (mean ± SD) (µg/mL) with dose of: | | | |
|---|---|---|---|---|
| | 320 mg/kg | 640 mg/kg | 1,280 mg/kg | 2,560 mg/kg |
| 5 | 530.3 ± 63 | 727.6 ± 87.4 | 1,353.8 ± 511 | 2,171 ± 659 |
| 10 | 646.4 ± 173 | 1,005.2 ± 229.5 | 1,542 ± 151 | 2,372 ± 374 |
| 15 | 506.2 ± 87.5 | 889.3 ± 45.7 | 1,720 ± 180 | 2,303 ± 249 |
| 30 | 333.3 ± 9.8 | 481 ± 43.1 | 953.1 ± 70 | 1,867 ± 248 |
| 60 | 151.2 ± 101.1 | 307.1 ± 51.8 | 472.3 ± 51.3 | 1,026 ± 233 |
| 120 | BLQ[a] | 191.9 ± 7.5 | 213.7 ± 34.5 | 392 ± 121 |

[a]BLQ, below the limit of quantification.

**TABLE 3** Amikacin plasma concentrations over time

| Time point (min) | Amikacin concentration (mean ± SD) (μg/mL) with dose of: | | | |
|---|---|---|---|---|
| | 320 mg/kg | 640 mg/kg | 960 mg/kg | 1,280 mg/kg |
| 5 | 145.97 ± 106.47 | 589.70 ± 195.16 | 630.96 ± 310.37 | 769.89 ± 307.99 |
| 10 | 43.31 ± 29.23 | 553.54 ± 149.92 | 938.60 ± 558.66 | 987.57 ± 496.23 |
| 15 | 110.85 ± 56.91 | 463.84 ± 52.67 | 695.31 ± 155.22 | 772.94 ± 326.85 |
| 30 | 26.94 ± 29.94 | 237.38 ± 142.52 | 472.03 ± 173.56 | 529.99 ± 238.09 |
| 60 | BLQ[a] | 97.20 ± 44.14 | 143.93 ± 57.97 | 340.95 ± 23.33 |
| 120 | BLQ | 25.92 | 20.80 ± 12.57 | 250.90 ± 29.55 |

[a]BLQ, below the limit of quantification.

nation and lung pathology with *P. aeruginosa* strains that are sensitive to the antibiotic but not bacterial strains that are resistant to the antibiotic.

**Validation of humanized doses of antibiotics by long-term survival.** Having validated that humanized doses of aztreonam and amikacin can reduce bacterial burdens in infected mice based on antibiotic resistance profiles, we next asked whether these regimens could protect mice against lethal infection by *P. aeruginosa* strains that are sensitive to the antibiotic. Equal numbers of male and female mice were infected by direct instillation into the lungs with 10× the $LD_{50}$ of *P. aeruginosa* strain 0230 (sensitive to aztreonam and resistant to amikacin) or strain 0231 (resistant to aztreonam and sensitive to amikacin). At 6 h postinfection, mice were administered 640 mg/kg aztreonam, 293 mg/kg amikacin, or vehicle only (PBS). Drug administration was repeated q6h for 5 days (120 h). At each administration, mouse temperatures were also measured. Mice were euthanized if they met predetermined endpoint criteria or at 5 days, and tissues were harvested for bacterial enumeration and histological analysis. As predicted by the significant reduction in bacterial burdens at 21 h postinfection (Fig. 5 and 6), mice receiving antibiotic did not display symptoms of infection and were protected from lethal infection, while mice receiving PBS all succumbed to infection within 48 h (Fig. 7 and 8). The only exception was one mouse in the amikacin group (Fig. 8). This mouse was also the only mouse that received antibiotics for which bacteria were recovered from tissues; bacterial burdens in all other antibiotic-treated animals were below our limit of detection. Minimal lung pathology was observed in antibiotic-

**TABLE 4** $T_{>MIC}$ for aztreonam in mice

| Dose | Dosing interval (h) | $T_{>MIC}$ (%) with MIC of: | | |
|---|---|---|---|---|
| | | 4 mg/L | 32 mg/L | >64 mg/L |
| Human dose | | | | |
| 2 g | 8 | 97 | 38 | 0 |
| | | | | |
| Mouse dose | | | | |
| 320 mg/kg | 4 | 51 | 21 | 0 |
| 640 mg/kg | 4 | 100 | 45 | 5 |
| 1,280 mg/kg | 4 | 96 | 50 | 19 |
| 2,560 mg/kg | 4 | 100 | 66 | 32 |
| 320 mg/kg | 6 | 34 | 14 | 0 |
| 640 mg/kg | 6 | 69 | 30 | 3 |
| 1,280 mg/kg | 6 | 64 | 33 | 12 |
| 2,560 mg/kg | 6 | 78 | 44 | 22 |
| 320 mg/kg | 8 | 26 | 10 | 0 |
| 640 mg/kg | 8 | 52 | 22 | 2 |
| 1,280 mg/kg | 8 | 48 | 25 | 9 |
| 2,560 mg/kg | 8 | 59 | 33 | 16 |
| 320 mg/kg | 12 | 17 | 7 | 0 |
| 640 mg/kg | 12 | 32 | 15 | 1 |
| 1,280 mg/kg | 12 | 35 | 17 | 6 |
| 2,560 mg/kg | 12 | 39 | 22 | 11 |

**TABLE 5** Calculated AUC$_{0-24}$ and $C_{max}$ values for amikacin

| Species and dosing interval (h) | Dose (mg/kg) | AUC$_{0-24}$ (mg · h/L) | $C_{max}$ (mg/L) |
|---|---|---|---|
| Human | | | |
| 24 | 15 | 370 | 65 |
| | | | |
| Mouse | | | |
| 24 | 605 | 370 | 541 |
| 24 | 142 | 20 | 65 |
| 12 | 385 | 370 | 236 |
| 12 | 142 | 40 | 65 |
| 6 | 293 | 370 | 109 |
| 6 | 142 | 80 | 65 |

treated animals, which was significantly less than that in PBS-treated animals ($P < 0.001$) (Fig. 7E and 8E), with the exception of the one animal that succumbed to infection in the amikacin-treated group. As predicted by the bacterial burden within the lungs, this animal showed pathology similar to that observed in the PBS-treated cohort (Fig. 8E). Together, these data demonstrate that the antibiotic regimens determined by PK studies protected mice from lethal respiratory disease by *P. aeruginosa* strains that are sensitive to the antibiotic.

## DISCUSSION

The rise of antibiotic resistance among bacterial pathogens is far outpacing the discovery of new antimicrobials to treat these infections. Efforts to identify and test novel antimicrobials need to be implemented to ensure that we can treat pathogens that become resistant to all current therapies. In addition to research into the discovery of new antimicrobials, the development of robust preclinical models that can rapidly screen novel INDs and accurately predict clinical success is important to speed the drug development pipeline. Moreover, validated preclinical animal models can be used to support clinical studies in which patient population sizes may be small due to infrequency of infection by specific MDR bacteria. Therefore, our goal in these studies was to improve upon an existing mouse model for the testing of antimicrobials against *P. aeruginosa* by expanding the number of MDR *P. aeruginosa* strains that have been characterized for virulence in this model, developing humanized regimens for two conventional antibiotics that can be used as comparators in subsequent IND screening, and validating that the model can accurately differentiate between successful and

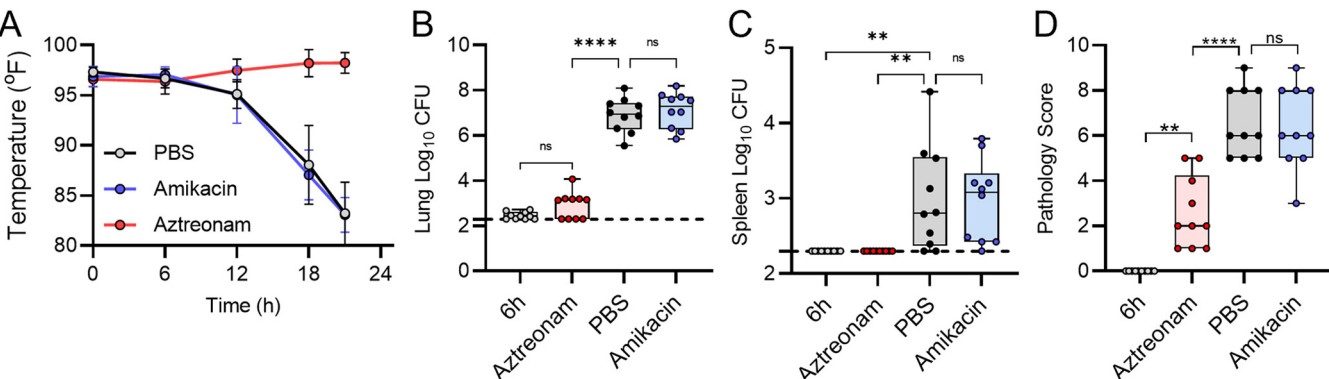

**FIG 5** Effects of antibiotic treatment on *P. aeruginosa* 0230 infection. Equal numbers of male and female mice ($n = 10$) were infected with 10× the LD$_{50}$ of *P. aeruginosa* 0230. At 6 h, animals were treated with PBS, aztreonam, or amikacin. (A) Temperatures of the animals were measured during the course of the experiment. (B and C) At 21 h, mice were euthanized, and bacterial numbers were enumerated from the lungs (B) or spleens (C). (D) Lung pathology was assessed at 21 h. Data for panels B to D were compared to those for samples harvested from a group of animals at 6 h (prior to treatment), and each circle represents data from an individual mouse. The dotted line indicates the limit of detection. **, $P \leq 0.01$; ****, $P \leq 0.0001$; ns, not significant, analysis of variance (ANOVA) with Tukey's test.

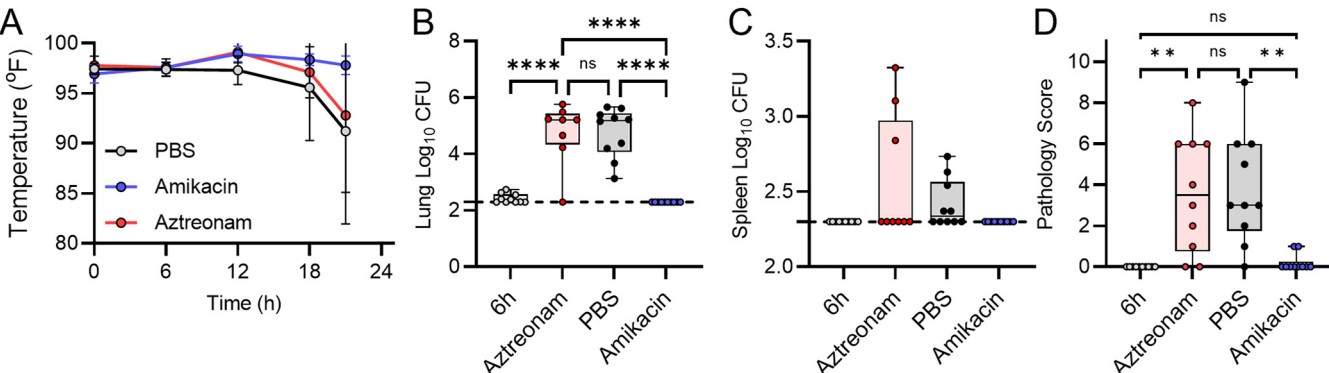

**FIG 6** Effects of antibiotic treatment on *P. aeruginosa* 0231 infection. Equal numbers of male and female mice (*n* = 10) were infected with 10× the LD$_{50}$ of *P. aeruginosa* 0231. At 6 h animals were treated with PBS, aztreonam, or amikacin. (A) Temperatures of the animals were measured during the course of the experiment. (B and C) At 21 h, mice were euthanized, and bacterial numbers were enumerated from the lungs (B) or spleens (C). (D) Lung pathology was assessed at 21 h. Data for panels B to D were compared to those for samples harvested from a group of animals at 6 h (prior to treatment), and each circle represents data from an individual mouse. The dotted line indicates the limit of detection. **, $P \leq 0.01$; ****, $P \leq 0.0001$; ns, not significant, ANOVA with Tukey's test.

unsuccessful treatments based on the inherent drug susceptibility of a given *P. aeruginosa* strain.

The model employed here is a transiently leukopenic mouse model that best recapitulates acute pulmonary infection of immunocompromised patients. We previously demonstrated that the induction of leukopenia significantly reduces the dose of bacteria required to establish a lethal infection (10). This reduction in infectious dose is important, because models requiring high doses of *P. aeruginosa* can artificially impact the interpretation of the efficacy of therapies, which we showed previously for meropenem using *P. aeruginosa* UNC-D. While treatment with meropenem could reduce bacterial burdens in an immunocompetent model, which required ~$10^8$ CFU to establish a lethal infection, the mice still succumbed to infection. In contrast, meropenem could reduce bacterial burdens and protect mice

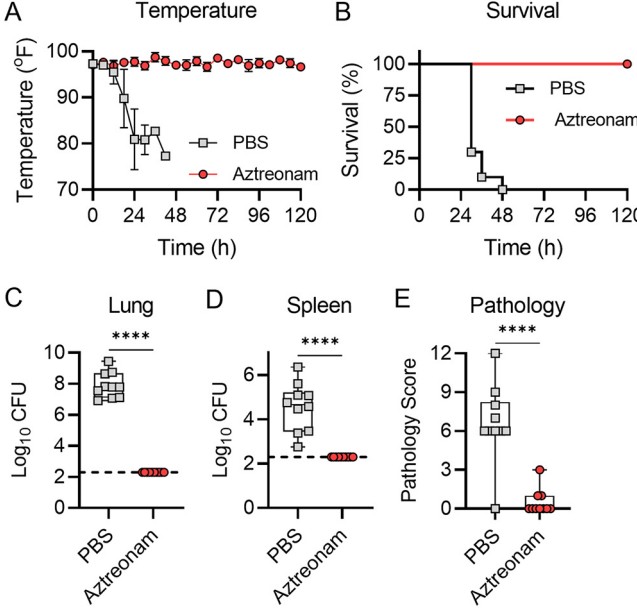

**FIG 7** Aztreonam efficacy during 5-day treatment. Equal numbers of male and female mice (*n* = 10) were infected with 10× the LD$_{50}$ of *P. aeruginosa* strain 0230. (A) Temperatures were measured at the indicated times. (B) Mouse survival. (C and D) Mice were euthanized if they reached predetermined endpoint criteria or at 5 days, and bacteria were enumerated in the lungs (C) or spleen (D). (E) A cross section of the lungs was excised and stained with H&E, and inflammation and pathology were assessed. For panels C to E, each symbol represents data from an individual mouse. ****, $P \leq 0.0001$, Student's *t* test.

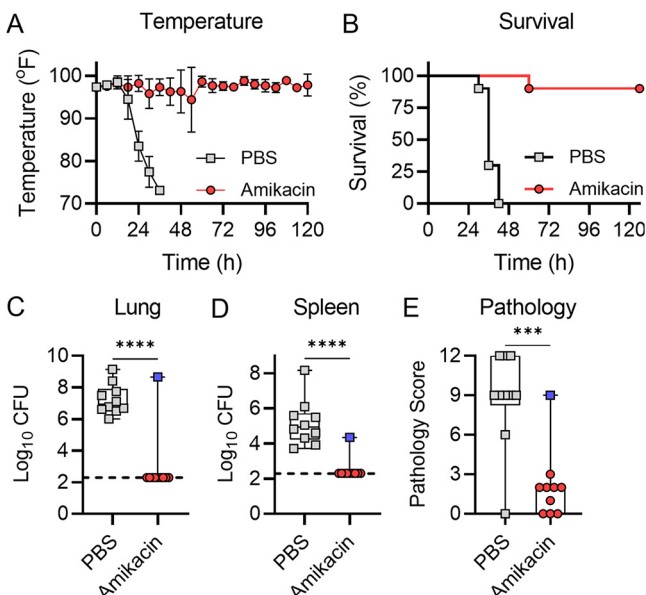

**FIG 8** Amikacin efficacy during 5-day treatment. Equal numbers of male and female mice (*n* = 10) were infected with 10× the LD$_{50}$ of *P. aeruginosa* strain 0231. (A) Temperatures were measured at the indicated times. (B) Mouse survival. (C and D) Mice were euthanized if they reached predetermined endpoint criteria or at 5 days, and bacteria were enumerated in the lungs (C) or spleen (D). (E) A cross section of the lungs was excised and stained with H&E, and inflammation and pathology were assessed. For panels C to E, each symbol represents data from an individual mouse. The blue squares in the amikacin group indicate the mouse that succumbed to infection in this group. ***, $P \leq 0.0001$; ****, $P \leq 0.0001$, Student's *t* test.

from lethal infection in the leukopenic model, which required only ~10$^5$ CFU to establish a lethal infection (10). However, the leukopenic model can still differentiate virulence between strains. We demonstrated this previously with two strains, i.e., the highly pathogenic laboratory-adapted strain PAO1 and a less virulent MDR clinical isolate, UNC-D (10), and have now independently confirmed this using four diverse MDR strains. While the reasons for these differences in virulence have not been defined and are outside the scope of these studies, these findings highlight the variety of pathogenicity among clinical isolates of *P. aeruginosa*. Moreover, we hope that these data raise awareness regarding the need to fully characterize individual strains in animals to ensure that proper infectious doses are used in subsequent *in vivo* testing of antimicrobials. Together, these two studies establish infection kinetics for six strains that are readily available through the NIH and CDC, and they increase the genetic diversity of the strains that can be used to test INDs in this preclinical model. This increased diversity, including diversity in known antibiotic resistance mechanisms, more closely represents the clinical populations and should provide a better prediction of IND success in the clinic.

The metabolism and physiology of mice differ from those of humans (20, 21). These differences can have significant effects on many classes of drugs, resulting in drugs that demonstrate promise in a mouse model but fail in clinical trials (22, 23). However, because antimicrobials directly target the pathogen and not the host, positive results in the mouse model have higher rates of successful translation into the clinic. Clinical success is further improved by incorporation of PK and PD analyses of antimicrobials in mice to identify driver indices that are independent of host metabolism (9). For this model, we chose the inbred BALB/c mouse line; one of the reasons was the assumption that an inbred mouse line with limited genetic diversity would demonstrate less variation in drug metabolism between individual animals. Less variation allows for smaller group sizes to achieve the statistical power required to identify appropriate PK/PD indices that govern efficacy. It is possible that outbred mice may have increased variance in drug metabolism that better reflects the human population. Because PK/PD indices for antimicrobials are independent of metabolism, this variation likely would not have a significant impact on the identification of the appropriate

PK/PD driver. However, if studies in outbred mice indicate that the PK/PD target has more variability in a genetically mixed population, then it could be informative in the subsequent design of dosing regimens for clinical studies with new INDs. Therefore, both inbred and outbred lines may provide useful insights to speed translation of novel antimicrobials to the clinic. Using aztreonam and amikacin, we established the feasibility of PK and PD analyses in leukopenic BALB/c mice during infection and were able to use these data to establish a dosing regimen that closely mimics the parameters achieved during clinical treatment of humans. Therefore, we have shown that PK analyses can be achieved with this model and applied during future characterization of INDs to establish the driver indices required for effective treatment during downstream clinical trials.

Finally, using aztreonam and amikacin, we were able to validate that the model could clearly differentiate between successful and unsuccessful outcomes, which were directly correlated with *in vitro* predictions of antibiotic susceptibility, as expected. These data confirm that *in vitro* MICs for INDs should reliably justify whether a novel antimicrobial be further evaluated in the mouse model. Moreover, the characterization of these two antibiotics provides robust comparators that can be included in future evaluations to help gauge the relative success of novel INDs. Ideally, INDs would perform as well as or better than these conventional antibiotics. Moreover, inclusion of aztreonam or amikacin during subsequent IND screening will provide data validating that the model is performing reproducibly during each use, eliminating potential errant results caused by unforeseen circumstances.

In summary, the data presented here demonstrate a robust preclinical mouse model that can be used to (i) generate data to justify subsequent studies in larger animal models and IND applications and (ii) provide additional data to support the smaller data sets that are usually generated in clinical studies targeting rarer MDR bacterial infections.

## MATERIALS AND METHODS

**Bacterial strains.** The *P. aeruginosa* strains used in these studies were provided by the CDC as part of the CDC and FDA Antibiotic Resistance Isolate Bank. These strains have been sequenced, and the antibiotic resistance profile of each strain has been fully characterized (Table 1). Bacteria were routinely cultured on Luria-Bertani (LB) agar and LB-Lennox broth. For infection, bacteria were recovered from cryopreservation on LB agar prior to inoculation of LB-Lennox broth for overnight growth at 37°C with aeration. Bacteria were washed into $1\times$ PBS to the desired concentration based on optical density at 600 nm ($OD_{600}$)-based estimates. All bacterial inocula were confirmed by serial dilution and enumeration on LB agar.

**Animals.** All animal studies were approved by the University of Louisville Institutional Animal Care and Use Committee (approval number 18368). Six-week-old male and female BALB/cJ mice (The Jackson Laboratory, Bar Harbor, ME) were acclimated on site for 8 days prior to bacterial challenge. Upon animal arrival, a temperature transponder (Bio Medic Data Systems, Seaford, DE) was implanted SC at the scruff of the neck. Neutropenia was induced in the animals via intraperitoneal injection of cyclophosphamide as described previously (10). Neutropenia (>90% depletion) was confirmed by complete blood cell counting 1 day prior to infection using a Hemavet 950 analyzer (Drew Scientific, Miami Lakes, FL). Group sizes for all studies were 10 animals, with 5 males and 5 females.

**Respiratory challenge with *P. aeruginosa*.** Mice were challenged with saline suspensions of *P. aeruginosa* by intubation-mediated intratracheal (IMIT) inoculation as described previously (10, 16). Animals were monitored for the development of moribund disease q8h after infection. Animals that met predetermined endpoint criteria (body temperature of ≤80.5°F or loss of righting reflex) were humanely euthanized by carbon dioxide asphyxiation and scored as succumbing to infection 8 h later.

**Natural history of *P. aeruginosa* infection.** Upon euthanasia, lungs and spleens were harvested. A representative section of lung was excised, fixed in 10% neutral buffered formalin for 24 h, and transferred into 70% ethanol prior to hematoxylin and eosin (H&E) staining. Stained sections were scored by a board-certified veterinary pathologist at the Iowa State University Veterinary Pathology Comparative Pathology Core and compared to naive control lungs. Pathology scoring was made on a four-point, four-criteria system, with a maximum score of 16 points. Tissues were scored in the areas of inflammation, infiltrate, necrosis, and other (including hemorrhage), and points were assigned as follows: no significant finding, 0; minimal pathology, 1; mild pathology, 2; moderate pathology, 3; severe pathology, 4. Bacteria were enumerated from remaining lung tissue and spleens as described previously (10).

**Defining simulated human doses of antibiotics.** Single-dose plasma PK studies were performed in infected mice. For each drug, four doses were utilized to generate a robust PK data set, from which a humanized dosing regimen was determined. Groups of four animals were administered single SC doses of antibiotic (amikacin, 320, 640, 960, and 1,280 mg/kg; aztreonam, 320, 640, 1,280, and 2,560 mg/kg). At six defined time points (5, 10, 15, 30, 60, and 120 min postadministration of the antibiotic), animals were euthanized, and plasma was collected from each mouse. Plasma samples were stored at −80°C until antibiotic quantification. Antibiotic concentrations in plasma were quantified and analytically assessed by HPLC with an Agilent Compact 1260 HPLC system as described previously (24, 25). Briefly, aztreonam

was analyzed with a XBridge BEH $C_{18}$ column and a VanGuard cartridge. An isocratic flow rate of 0.5 mL/min for the eluent (0.005 M tetrabutylammonium hydrogen sulfate [pH 3.0]-acetonitrile [88:12, by volume]) and UV detection at 280 nm were used for the analysis. The validation range and performance were recorded as follows: lower limit of quantification [LLOQ], 5 $\mu$g/mL; upper limit of quantification [ULOQ], 250 $\mu$g/mL; quality control [QC], 25 $\mu$g/mL; precision, 96.36%; accuracy, 88.26%. Amikacin was conjugated with 4-chloro-3,5-dinitrobenzotrifluoride (CNBF) before analysis. The resulting amikacin-CNBF was analyzed with an Agilent Eclipse Plus $C_{18}$ column. A gradient of methanol and deionized water (with 0.1% trifluoroacetic acid [TFA]) was used; the proportion of methanol was increased from 35% to 60% in the initial 2 min, kept at 60% for 1.5 min, increased to 90% in the next 1 min, and kept at 90% for another 1 min. The proportion of methanol was then reduced to 35% in 2 min and kept at 35% for another 1.5 min. A flow rate of 1.6 mL/min and UV detection at 238 nm were used for the analysis. The validation range and performance were recorded as follows: LLOQ, 71 $\mu$g/mL; ULOQ, 1,428 $\mu$g/mL; QC, 357 $\mu$g/mL; precision, 94.98%; accuracy, 95.59%.

PK parameters (means $\pm$ standard deviations [SDs]) were calculated using a time-ordered data set generated from the pooled PK data. Elimination half-life ($t_{1/2}$), $AUC_{0-\infty}$, and $C_{max}$ were calculated using a noncompartmental model with mean concentration values from each group of mice. The $t_{1/2}$ was determined by linear least-squares regression. The AUC was calculated from the mean concentrations using the trapezoidal rule. PK estimates for dose levels that were not directly measured were calculated using linear interpolation for dose levels between those with measured kinetics and linear extrapolation for dose levels above or below the highest and lowest dose levels with kinetic measurements.

Human PK data for aztreonam and amikacin were utilized to generate simulated human drug concentration-time exposure profiles for aztreonam administered intravenously at 2 g q8h and amikacin administered at 15 mg/kg once daily (17–19). Using the PK results generated above, we modeled dosing regimens in the mice that would be expected to provide drug concentration-time exposures similar to those in humans. We utilized the PK/PD parameter $T_{>MIC}$ for aztreonam and evaluated both $C_{max}$/MIC and $AUC_{0-24}$/MIC for amikacin (26).

**Evaluation of antibiotic efficacy. (i) Reduction in bacterial burdens.** Six hours after IMIT instillation of *P. aeruginosa* 0230 or 0231 (10$\times$ the $LD_{50}$), mice were administered humanized doses of either amikacin (293 mg/kg q6h) or aztreonam (640 mg/kg q6h). Animal temperatures were measured at every treatment. Twenty-one hours postinfection, mice were euthanized, and bacterial numbers were determined in the lungs and spleen by serial dilution and enumeration on agar plates. A section of the lungs was also processed for histological analysis as described above.

**(ii) Overall survival rates.** Six hours after IMIT instillation of *P. aeruginosa* 0230 or 0231 (10$\times$ the $LD_{50}$), mice were administered humanized doses of either amikacin (293 mg/kg q6h) or aztreonam (640 mg/kg q6h) for 5 days. Animal temperatures were measured at every treatment. Blood, lungs, and spleens were harvested from moribund animals or at 5 days postinfection from animals that survived to the end of the study. Bacterial numbers in these tissues were determined by serial dilution and enumeration on agar plates. A section of the lungs was also processed for histological analysis as described above.

**Statistical analysis.** Kaplan-Meier survival curves were fit to the data for each bacterial strain and the median survival time noted by gender and overall. The $LD_{50}$ values were calculated by fitting the logistic model (% dead = $\log_{10}$[dose]) and then calculating the dose at which 50% of the mice would be predicted to survive. To assess the natural history of infection, a repeated-measures linear model was fit to temperatures according to bacterial strain, assuming an autoregressive correlation structure. Power calculations estimated the significant lung burden ($\log_{10}$ CFU) to be seen for 80% power and sample sizes of 10 mice per treatment groups, assuming an alpha value of 0.05 and a possible 20% increase in variability within the treatment group. The disease burdens in the lungs and spleen were modeled on the $\log_{10}$ scale with sex and hour as the independent variables. Fisher's exact tests compared the incidence of pathology. All data were analyzed by sex and in aggregate. No significant differences by sex were observed.

## ACKNOWLEDGMENTS

This work was supported by funding from the FDA under contract HHSF223201810171C (M.B.L.).

We thank members of the FDA advisory panel for their advice during these studies.

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
