## [Reviewer comments · Microbiology Spectrum]

Microbiology Spectrum

Validated Preclinical Murine Model for Therapeutic Testing against Multidrug Resistant *Pseudomonas aeruginosa*

Jonathan Warawa, Xiaoxian Duan, Charles Anderson, Julie Sotsky, Daniel Cramer, Tia Pfeffer, Haixun Guo, Robert Adcock, Alexander Lepak, David Andes, Stacey Slone, Arnold Stromberg, Jon Gabbard, William Severson, and Matthew Lawrenz

Corresponding Author(s): Matthew Lawrenz, University of Louisville

Review Timeline:

Submission Date:	July 13, 2022
Editorial Decision:	August 10, 2022
Revision Received:	August 24, 2022
Accepted:	August 24, 2022

Editor: Jennifer Gaddy

Reviewer(s): Disclosure of reviewer identity is with reference to reviewer comments included in decision letter(s). The following individuals involved in review of your submission have agreed to reveal their identity: Neal D. Hammer (Reviewer #1)

Transaction Report:

DOI: <https://doi.org/10.1128/spectrum.02693-22>

August 10, 2022

Dr. Matthew B Lawrenz
University of Louisville
Department of Microbiology and Immunology
CTRB 618
505 S. Hancock
Louisville, KY 40202

Re: Spectrum02693-22 (Validated Preclinical Murine Model for Therapeutic Testing against Multidrug Resistant *Pseudomonas aeruginosa*)

Dear Dr. Matthew B Lawrenz:

I hope this email finds you doing well. I'm pleased to inform you that your manuscript was received favorably by peer Reviewers. Overall the Reviewers found your manuscript to be well-written and the data presentation to be easy to follow. The Reviewers had several minor text changes and a few questions that required clarification or modification of your tables and figures. We hope that you will consider making these changes to the manuscript and resubmitting a modified draft!

Link Not Available

Please do not hesitate to reach out if you need any further clarification or assistance.

Warmly,

Jennifer Gaddy, Ph.D.

Journals Department
Reviewer comments:

Reviewer #1 (Comments for the Author):

'Validated Preclinical Murine Model for Therapeutic Testing against multidrug resistant *Pseudomonas aeruginosa*' advances a respiratory mouse model of *Pseudomonas aeruginosa* infection by reporting the virulence for four different strains and

quantifying the efficacy of aztreonam and amikacin towards responsive and recalcitrant infection. Expanding the data set to include several multidrug resistant strains and antibiotic effectiveness addresses a considerable knowledge gap and provides a baseline for investigators to compare their investigational drug of choice. In the era of increasing antibiotic resistance, the study represents an important tool for development of new therapeutic strategies. The authors build on their previous work that established a murine lung disease model of *Pseudomonas aeruginosa* for therapeutic testing. This manuscript determines the LD50 for four multidrug resistant strains (0230, 0231, 0241, and 0246) and tracks bacterial colonization and dissemination by quantifying burdens in the lungs and spleen, respectively, at 3, 6, 12, and 21 h post infection. Pathobiology of infection was tracked over time by measuring temperature and inflammation via H&E staining. After establishing a virulence baseline for each strain, Warawa et al. sought to define aztreonam (azt) and amikacin (ami) as comparator antibiotics. Serum levels of the antibiotics after 5, 10, 15, 30, 60, and 120 min of administration were tracked and compared across increasing doses. Results indicated a regimen of 640 mg/kg/6h for azt and 293 mg/kg/6h for ami. These parameters were validated against responsive and recalcitrant strains. Specifically, strain 0230 was used to test efficacy of azt (ami served as a resistant comparator), while 0231 was used to determine ami efficacy (azt served as the resistant comparator). In both instances, the humanized regimen administered at 6 h post infection was determined to be effective via quantification of infection (lung burdens at 21), dissemination (spleen burdens at 21 h), lung pathology (at 21 hr), and temperature at 6 h, 12 h, 18 h, and 24h. This is a straight-forward, well-crafted study that is of broad interest to researchers developing new antibiotics. The conclusions are consistent with the data and the study provides important baseline for the development of novel, anti-pseudomonas drugs. The following comments are offered to improve the manuscript.

To make the data easier to find, the LD50 results could probably be included in the last column of Table 1.

Line 197 - *P. aeruginosa* should be italicized.

Reviewer #2 (Comments for the Author):

Summary:

This paper validates the effectiveness of a murine model of *P. aeruginosa* infection for following infection course and determining antibiotic efficacy. The strength of this leukopenic mouse model is that it does not require the extremely high infectious dose that other models require, which has the potential to perturb assessment of efficacious doses of antibiotics.

Major points to address:

- Line 61: To me, it seems like the use of inbred mouse strains to limit variation is a strike against the use of mouse strains rather than a mark in favor of the use of mouse strains to investigate the use of drugs in preclinical settings. After all, these data are supposed to translate to use in humans which have a lot of genetic variation relative to inbred mice. It seems that we want to use preclinical tests that will provide powerful data despite inherent genetic variability. Can you speak to this in your paper? Or maybe propose in the conclusions section that models using outbred mice could also be tested to enable the better preclinical identification of promising results despite inherent host genetic variation?
- Lines 65-69 need clarification. Are you saying that testing the specific drug in mice is better than comparison to similar drugs in humans to determine efficacy?
- Line 93, website citation should not just be a link - there are more appropriate ways to cite websites in text for publication.
- Lines 146-147, you state the justification for choosing aztreonam and amikacin was that the 4 strains you used have different susceptibility. I am curious if there is varying susceptibility amongst other *P. aeruginosa* strains. You state that you hope to use this model for comparison if novel drugs will work, but I have concerns about how applicable this comparison method will be if there is no differing susceptibility amongst other strains. Would the only strains you are able to compare be the ones with similar susceptibility patterns? That would significantly limit the useful scope of this model.
- Line 160: What is the rationale for subcutaneous drug delivery when it is stated in line 159 that humans receive this drug intravenously. It is probably better to use drug delivery routes more akin to human treatments for your mouse model.
- Line 107; >50% mortality achieved at which time point?
- Lines 256-260 are better suited for the introduction rather than your discussion. But again, the emphasis on needing models that provide useful data independent of host metabolism to me indicates a strong need for a model with genetic variation rather than an inbred model.
- Figure 6D, the red and blue colors appear to be switched from what you have been showing.
- Line 320, fig 4; '(Amikacin 320, 640, 960, and 1280 mg/kg; Aztreonam 320, 640, 1280, 2560 mg/kg)'. How were these doses selected?
- Line 361; Considering the limited resources and number of preclinical candidates, it is not always possible to estimate the humanized does of any preclinical candidate compounds. In that case, it would be great if the author suggested a method for selecting drug concentration for the in vivo study based on in vitro drug MIC and selectivity index (SI).

Minor points to address:

- I think it would make the sentence in lines 46-49 more powerful if you had a citation(s) directly after it.
- Line 53 you need a space before your bracket around the citation.
- Lines 56-63 would benefit from a citation on why mouse models are so useful/beneficial.
- Line 88, should be "previously."
- Line 130, should be "significantly."
- Line 185, "began developing signs."

- Line 184, 196: Italicize all instances of "*P. aeruginosa*".
- Line 34; Please provide the full form of LD50
- Line 48; Please provide the full form of CDC
- Line 74; Can you replace the word 'recapitulates' with 'mimics'
- Line 92; Please provide the full form of FDA
- Line 102; Since it requires special skill to infect mice with direct installation (intratracheal inoculation), is it possible to achieve the infection model through intranasal (inhalation) route? An ideal model should be "user friendly".
- Line 171-173; Is there any difference in effective humanized dose from male to female?
- Line 184, 270; Please italicize 'in vitro'
- Table 1; What is Inter.? I assumed "intermediate" but this should be clarified. Also, the full form of VIM, KPC, IMP and NDM should be added in the legend.
- Fig 5 & 6; 'Affect' can be replaced with 'Effect'

Staff Comments:

Preparing Revision Guidelines

Please return the manuscript within 60 days; if you cannot complete the modification within this time period, please contact me. If you do not wish to modify the manuscript and prefer to submit it to another journal, please notify me of your decision immediately so that the manuscript may be formally withdrawn from consideration by Microbiology Spectrum.

The authors would like to thank the reviewers for their time and constructive evaluations of the manuscript. We have tried to answer their questions and incorporate all their suggestions into the manuscript when appropriate. The following is a point-by-point response to each critique.

Reviewer #1 (Comments for the Author):

The following comments are offered to improve the manuscript.

1. To make the data easier to find, the LD50 results could probably be included in the last column of Table 1.
2. Line 197 - *P. aeruginosa* should be italicized.

Response: We made both of these modifications as recommended.

Reviewer #2 (Comments for the Author):

Major points to address:

1. Line 61: To me, it seems like the use of inbred mouse strains to limit variation is a strike against the use of mouse strains rather than a mark in favor of the use of mouse strains to investigate the use of drugs in preclinical settings. After all, these data are supposed to translate to use in humans which have a lot of genetic variation relative to inbred mice. It seems that we want to use preclinical tests that will provide powerful data despite inherent genetic variability. Can you speak to this in your paper? Or maybe propose in the conclusions section that models using outbred mice could also be tested to enable the better preclinical identification of promising results despite inherent host genetic variation?

Response: Thank you for your comments about outbred mice. We believe that inbred and outbred mice both have a place in preclinical screening of antibiotics. For us, the BALB/c line is a well-established model for *P. aeruginosa* pulmonary infection, making it a robust infection model for testing drugs for MDR *P. aeruginosa*. We also assumed that limited genetic diversity would result in less variation in drug metabolism, which will allow us to use smaller group sizes when identifying PK/PD drivers for future INDs. It is possible that outbred mice may demonstrate greater variance in the metabolism of individual drugs, which may better reflect what will be seen in the human population. This variation will not have a major impact on the identification of the PK/PD driver because for antimicrobials these indices are independent of host metabolism, but it could be informative in the subsequent design of dosing regimens for clinical studies to account for potential greater variation in drug metabolism in the human population. We have added a section to the discussion to address these ideas (Lines 263-273).

2. Lines 65-69 need clarification. Are you saying that testing the specific drug in mice is better than comparison to similar drugs in humans to determine efficacy?

Response: No, we are saying that because PK/PD drivers for antimicrobials are independent of host metabolism, the mouse model is a better predictor of success for antimicrobial than it has been for other classes of drugs that are not antimicrobials. We modified the sentence to better clarify this.

3. Line 93, website citation should not just be a link - there are more appropriate ways to cite websites in text for publication.

Response: Moved to references as recommended.

4. Lines 146-147, you state the justification for choosing aztreonam and amikacin was that the 4 strains you used have different susceptibility. I am curious if there is varying susceptibility amongst other *P. aeruginosa* strains. You state that you hope to use this model for comparison if novel drugs will work, but I have concerns about how applicable this comparison method will be if there is no differing susceptibility amongst other strains. Would the only strains you are able to compare be the ones with similar susceptibility patterns? That would significantly limit the useful scope of this model.

Response: There is likely broad variation in susceptibility to these antibiotics. However, for the purpose of these studies, these differences in susceptibility of the panel we chose allowed us to validate that the mouse model accurately recapitulates antibiotic success and failure scenarios as predicted by the *in vitro* MIC susceptibility to these antibiotics. In other words, this panel allowed us to use amikacin and aztreonam to demonstrate that the model should be predictive in estimating success for novel therapeutics based on *in vitro* MIC data. If using these strains, amikacin and aztreonam can be used as a comparator for affective treatment. This does not preclude the use of other strains within the model, but additional strains would need to be characterized for

virulence (LD50 and natural history) prior to testing. Limitations would come from virulence of the strain, not on individual antibiotic resistance profiles. If comparator antibiotics are required, then susceptibility profiles of the new panel would have to be evaluated to identify an appropriate antibiotic for comparison.

5. Line 160: What is the rationale for subcutaneous drug delivery when it is stated in line 159 that humans receive this drug intravenously. It is probably better to use drug delivery routes more akin to human treatments for your mouse model.

Response: Many antibiotics use IV infusion in the clinic, but this is difficult to recapitulate in the mouse model, especially in a cost-effective manner. While single bolus IV injections can be performed, this is a labor-intensive procedure, especially if multiple injections are required. However, IV injection is not required if other routes of administration allow for diffusion of the drug into the blood/site of infection. Because subcutaneous delivery of these two antibiotics results in drug diffusion into the blood, and we could meet the PK/PD indices in the plasma that govern the efficacy of the antibiotic via the subcutaneous route, this is an appropriate route for administering these two antibiotics. However, this does not preclude other routes of administration for future testing of novel antimicrobials in this model. Routes would be chosen based on the PK/biodistribution requirements for each drug. Importantly, the PK/PD indices driving efficacy are not dependent on the route of administration, only on the concentration of drug in the blood (in these cases), and so the route of administration in the mouse does not change what parameters need to be met in humans and does not indicate how the drug can/should be administered to humans. While IV administration might change the dosing schedule in the mouse, it will not change the PK/PD indices that need to be met in mouse or human, and so routes other than IV administration in the mouse are appropriate for preclinical testing of antibiotics.

6. Line 107; >50% mortality achieved at which time point?

Response: Added “within 50 h post-infection” line 107.

7. Lines 256-260 are better suited for the introduction rather than your discussion. But again, the emphasis on needing models that provide useful data independent of host metabolism to me indicates a strong need for a model with genetic variation rather than an inbred model.

Response: We introduce this idea in the introduction (lines 66-70) but believe we need to include this in the discussion, especial with the new inclusion of the discussion of outbred animals as suggested by the reviewer.

8. Figure 6D, the red and blue colors appear to be switched from what you have been showing.

Response: Thank you for catching this. It is now fixed.

9. Line 320, fig 4; '(Amikacin 320, 640, 960, and 1280 mg/kg; Aztreonam 320, 640, 1280, 2560 mg/kg)'. How were these doses selected?

Response: These were chosen based on the literature, toxicity, and our ability to detect the drug in plasma.

10. Line 361; Considering the limited resources and number of preclinical candidates, it is not always possible to estimate the humanized does of any preclinical candidate compounds. In that case, it would be great if the author suggested a method for selecting drug concentration for the in vivo study based on in vitro drug MIC and selectivity index (SI).

Response: We are not sure the goal of the adding this to the methods? We are also confused by the comment of “it is not always possible to estimate the humanized dose for a preclinical candidate”. For this manuscript, the use of phrase “humanized dose” refers to a dose that mimics that PK/PD drivers that are met in humans with the current treatment regimen used in the clinic. However, for a novel drug, we would not have this data or a humanized dose. Instead, we would propose to do a PK/PD study to identify the indices that drive efficacy of the IND. We agree that the selective index and MIC could be used to help define potential dosing regimens that could be used in the model, but these considerations seem outside the scope of the current manuscript.

Minor points to address:

1. Lines 56-63 would benefit from a citation on why mouse models are so useful/beneficial.

Response: We are not aware of a reference that describes why the mouse model is the most widely used model, but we identify three beneficial reasons that support the wide use of the mouse model. If the reviewer has a specific reference in mind, we are happy to review and add if appropriate.

2. Line 102; Since it requires special skill to infect mice with direct installation (intratracheal inoculation), is it possible to achieve the infection model through intranasal (inhalation) route? An ideal model should be "user friendly".

Response: While it might be possible, the infectious dose will be significantly impacted by intranasal administration (it will be significantly higher as only about 10% of the inoculum makes it to the lungs). If a user wanted to use this route, they would have to reestablish the LD50 and natural history for each strain.

3. Line 171-173; Is there any difference in effective humanized dose from male to female?

Response: All groups contained male and female mice and we did not observed differences in the PK data or in efficacy when the humanized dose between the two sexes.

4. Table 1; What is Inter.? I assume "intermediate" but this should be clarified. Also, the full form of VIM, KPC, IMP and NDM should be added in the legend.

Response: We added intermediate and clarified that Known Resistance refers to specific genes.

5. I think it would make the sentence in lines 46-49 more powerful if you had a citation(s) directly after it.

6. Line 53 you need a space before your bracket around the citation.

7. Line 88, should be "previously."

8. Line 130, should be "significantly."

9. Line 185, "began developing signs."

10. Line 184, 196: Italicize all instances of "*P. aeruginosa*".

11. Line 34; Please provide the full form of LD50

12. Line 48; Please provide the full form of CDC

13. Line 74; Can you replace the word 'recapitulates' with 'mimics'

14. Line 92; Please provide the full form of FDA

15. Line 184, 270; Please italicize 'in vitro'

16. Fig 5 & 6; 'Affect' can be replaced with 'Effect'.

Response: for points 5-16, we made all the changes as suggested by the reviewer.

August 24, 2022

Dr. Matthew B Lawrenz
University of Louisville
Department of Microbiology and Immunology
CTRB 618
505 S. Hancock
Louisville, KY 40202

Re: Spectrum02693-22R1 (Validated Preclinical Murine Model for Therapeutic Testing against Multidrug Resistant *Pseudomonas aeruginosa*)

Dear Dr. Matthew B Lawrenz:

We are pleased to inform you that your manuscript has been accepted, and I am forwarding it to the ASM Journals Department for publication. You will be notified when your proofs are ready to be viewed.

Sincerely,

Jennifer Gaddy
Editor, Microbiology Spectrum
